# Evaluation of Pathogenicity and Causativity of Variants in the *MPZ* and *SH3TC2* Genes in a Family Case of Hereditary Peripheral Neuropathy

**DOI:** 10.3390/ijms24129786

**Published:** 2023-06-06

**Authors:** Olga Shchagina, Mariya Orlova, Aisylu Murtazina, Alexandra Filatova, Mikhail Skoblov, Elena Dadali

**Affiliations:** Research Centre for Medical Genetics, Moscow 115522, Russia

**Keywords:** CMT, *SH3TC2*, *MPZ*, HMSN, gene panel, splicing variants, minigene assay

## Abstract

The implementation of NGS methods into clinical practice allowed researchers effectively to establish the molecular cause of a disorder in cases of a genetically heterogeneous pathology. In cases of several potentially causative variants, we need additional analysis that can help in choosing a proper causative variant. In the current study, we described a family case of hereditary motor and sensory neuropathy (HMSN) type 1 (Charcot–Marie–Tooth disease). DNA analysis revealed two variants in the *SH3TC2* gene (c.279G>A and c.1177+5G>A), as well as a previously described variant c.449−9C>T in the *MPZ* gene, in a heterozygous state. This family segregation study was incomplete because of the proband’s father's unavailability. To evaluate the variants’ pathogenicity, minigene splicing assay was carried out. This study showed no effect of the *MPZ* variant on splicing, but the c.1177+5G>A variant in the *SH3TC2* gene leads to the retention of 122 nucleotides from intron 10 in the RNA sequence, causing a frameshift and an occurrence of a premature stop codon (NP_078853.2:p.Ala393GlyfsTer2).

## 1. Introduction

Hereditary motor and sensory neuropathy (HMSN), or Charcot–Marie–Tooth disease (CMT), is a disorder of the peripheral nerves predominantly characterized by distal atrophy and sensory impairment. In addition, patients may also have hand and foot deformities and ataxia. Some patients have additional symptoms, such as spinal deformities and cranial nerve involvement [1]. Nerve conduction studies of patients show signs of peripheral nerve pathology, with or without nerve conduction velocity (NCV) slowing. In the case of a decrease in median NCV below 38 m/s, demyelinating neuropathy or HMSN type 1 is assumed.

All HMSNs are a highly genetically heterogeneous group of hereditary disorders. HMSN type 1 may be associated with pathogenic variants in genes encoding structural components of the myelin sheath (*PMP22*, *MPZ*, *NDRG1*), proteins mediating and regulating the exchange of ions and small metabolites (*GJB1*), or proteins responsible for the neuron transport system (*NEFL*), for signal transfer inside of the neuron (*LITAF*, *SH3TC2*), or for proliferation (*EGR2*).

It is quite difficult in clinical practice to suggest a gene that might have pathogenic variants causing peripheral neuropathy without molecular tests. Some distinct clinical signs are associated with certain neuropathy-like stroke-like episodes and white matter alterations in CMT1X (*GJB1*), early scoliosis in CMT4C (*SH3TC2*), or neuromyotonic discharges recorded by electromyography in axonal neuropathy with neuromyotonia (*HINT1*) [2,3]. Nevertheless, despite all these helpful clinical correlations, diagnostic strategies are being replaced by a targeted next-generation sequencing (NGS) as the most effective and economical approach. According to the international Inherited Neuropathy Consortium (INC), 997 (60.4%) out of 1652 patients from 13 centers had a genetically confirmed diagnosis [4].

According to the data presented in the ORPHANET portal (https://www.orpha.net/ accessed on 1 June 2023) [5], the majority of laboratories with available CMT analysis use a complex approach of molecular studies including quantitative analysis (Multiplex Ligation-dependent Probe Amplification (MLPA) or another), Sanger sequencing of target genes, next-generation sequencing (NGS) using panels of varying sizes (60 to 100 genes), or whole-exome/genome sequencing (WES/WGS).

WES/WGS may be carried out for patients with a cause of the disease undetected by targeted methods. However, during genetic screening, it is crucial not to overestimate variants of unknown significance (VUS) as causative [6,7].

The complication of using methods allowing us to analyze multiple genes is the detection of several variants in different genes that could lead to HMSN and be pathogenic. The majority of these variants are classified as VUS according to the ACMG criteria [8] and require further examination to evaluate their pathogenicity and causativity. In the current study, we evaluated the pathogenicity and causativity of the following three variants: *SH3TC2* c.279G>A and c.1177+5G>A and *MPZ* c.449−9C>T. These variants were identified in a CMT1 patient and his affected sister during DNA diagnostics. We have conducted a family segregation analysis, a functional analysis using the minigene construct method, and an evaluation of symptom specificities for this purpose.

## 2. Results

The proband was a 32-year-old male patient referred to the Research Centre for Medical Genetics with a diagnosis of hereditary neuropathy. From the age of 10 years, he often stumbled and fell. At the age of 26 years, the patient began to notice a change in his gait, but he did not pay attention to these symptoms. When he was applying for a job and undergoing a medical examination at the age of 30 years, the neurologist found signs of neuropathy and referred the patient to a geneticist. Neurological examination showed moderate hypotrophy of the hand muscles, severe atrophy of the feet and lower leg muscles, foot drop, and steppage gait (Figure 1B). The proband had prominent thoracic kyphoscoliosis, hand muscle weakness (4/5), lower leg and foot muscle weakness (1/5), hand joint contractures, hyporeflexia of upper limbs, and areflexia of lower limbs. He could not stand on his heels and toes (Figure 1B). Pain sensations were decreased in distal parts of the limbs, and vibration sense was absent in the legs.

A 25-year-old sister of the proband (Figure 1A) had complaints of episodic dizziness and unsteadiness. Neurological examination showed mild hypotrophy of hand and foot muscles and weakness in extension of the thumbs (4/5) (Figure 1C). She had no sensation disturbances, foot deformities, or scoliosis.

Both siblings underwent a nerve conduction examination. In the proband, it revealed signs of demyelination of the motor and sensory nerves of the upper and lower limbs. Sensor nerve action potentials (SNAPs) were absent in the legs, as were compound muscle action potentials (CMAPs) from peroneal nerves. CMAPs from tibial nerves had decreased amplitude and severe temporal dispersion. Tibial conduction velocity (CV) was 15 m/s (N > 40 m/s); the median CV was 28 m/s (N > 50 m/s). In the proband’s sister, SNAPs from the sural nerve were also absent. Tibial and peroneal CMAPs had decreased amplitude and increased duration; CV was also decreased as in her brother to 23–28 m/s (N > 40 m/s).

The proband’s mother had no clinical or electrophysiological signs of peripheral neuropathy. The proband’s father died of an unknown cause at an age over 40. Relatives did not notice any signs of neuropathy in him.

After excluding a duplication of the *PMP22* gene using quantitative MLPA, we carried out molecular genetic analysis with a target NGS panel consisting of 21 causative genes for HMSN and detected three variants showing signs of pathogenicity in both patients.

Panel sequencing was performed for the affected siblings. Both brother and sister had three heterozygous variants identified: *SH3TC2* (NM_024577.4): c.279G>A (p.Lys93=), *SH3TC2* (NM_024577.4): c.1177+5G>A, and *MPZ* (NM_000530.8): c.449−9C>T. Later, we performed Sanger sequencing on the healthy mother. She was found heterozygous for the SH3TC2 gene, carrying one normal allele and the variant SH3TC2 (NM_024577.4): c.1177+5G>A. The results of the study of the mother (the father of the patients is unavailable for analysis) suggest a compound-heterozygous state of *SH3TC2* variants in siblings.

The c.449−9C>T variant in the *MPZ* gene (NM_000530.8) was previously described in a heterozygous state in a 46-year-old male Serbian patient with neural amyotrophy [9], the only affected family member with the age of manifestation around 26 years, with distal limb muscle atrophy, decreased tendon reflexes, and a median nerve conduction velocity of 22.7 m/s. Aside from that, the patient had pes cavus, scoliosis, tremor, and sensory ataxia. His healthy parents were not examined; therefore, the researchers were not sure if the detected variant was causative.

Aside from this variant, we detected two variants in the *SH3TC2* gene (NM_024577.4): c.279G>A (p.Lys93=) and c.1177+5G>A. The synonymous substitution c.279G>A was previously described in two patients from the Czech Republic [10]. In one of them, it was detected in a compound heterozygous state with a pathogenic variant p.Arg954Stop and, in the other, with p.Tyr169His. Both patients were the only affected members of their families. HMSN manifested in the first patient, a 43-year-old male, at the age of 11 years; he had noticeable atrophy of distal limb muscles, sensitivity impairments, pes cavus, and NCV corresponding to demyelinating neuropathy. The second patient, a 53-year-old female with the manifestation age of 12 years, had the same symptoms but more pronounced sensitivity impairments in the distal lower limb muscles. Later, functional analysis for this synonymous variant confirmed its effect on splicing [11].

The c.1177+5G>A variant was mentioned once in the ClinVar database as VUS in a patient with CMT4 [12]. In one study, a variant leading to the substitution of the next nucleotide—c.1177+6 T>C—was described as a splice site mutation in a compound heterozygous state with E553* leading to a CMT4C phenotype; however, no proof of this variant’s pathogenicity was presented [13]. Mutations in the *SH3TC2* gene cause an autosomal recessive form of CMT4C, which manifests during the second decade of life and is characterized by pronounced scoliosis [14].

The segregation analysis showed the presence of all three variants: *MPZ* (NM_000530.8): c.449−9C>T, *SH3TC2* (NM_024577.4): c.279G>A (p.Lys93=), and *SH3TC2* (NM_024577.4): c.1177+5G>A in both patients’ genotypes. Their healthy mother only had the c.1177+5G>A variant in the *SH3TC2* gene, which allowed us to assume that the variants in the patients are located on different chromosomes.

In the GnomAD [15] database, the frequency of the *MPZ* (NM_000530.8): c.449−9C>T variant is 0.00004469 in non-Finnish Europeans in a heterozygous state (on five alleles out of 111,880) and, in the control cohort of patients without neural pathology, on one out of 87,768 alleles in a person above the age of 80 years. In 2090 exomes of Russian patients with various hereditary pathologies, this variant was not detected [16].

The *SH3TC2* (NM_024577.4): c.279G>A (p.Lys93=) variant was encountered in GnomAD in a heterozygous state on five chromosomes of non-Finnish Europeans out of 120,032 (0.00003875), and on three out of 103,156 chromosomes in the control cohort, and was not detected in the 2090 exomes of Russian patients with various hereditary pathologies. The *SH3TC2* (NM_024577.4): c.1177+5G>A variant was not registered in GnomAD or in the 2090 exomes of Russian patients with various hereditary pathologies.

According to the Human Splicing Finder [17], the *MPZ* (NM_000530.8): c.449−9C>T variant does not affect splicing, and both of the *SH3TC2* (NM_024577.4) variants affect the donor splice sites. SpliceAI also shows the absence of effect of *MPZ* (NM_000530.8): c.449−9C>T on splicing. The *SH3TC*2 (NM_024577.4): c.279G>A (p.Lys93=) variant affects the donor splice site (Donor Loss Δ score 0.59; Donor Gain—0.73), which was confirmed by functional analysis [11], and *SH3TC2* (NM_024577.4): c.1177+5G>A leads to the loss of the donor splice site (Donor Loss Δ score 0.64).

To determine the effect of the intronic variant c.449−9C>T on *MPZ* pre-mRNA splicing, we used the minigene splicing assay. For this process, we cloned MPZ exon 4 with flanking intronic sequences into the intron of the pSpl3-Flu2 vector. The transcripts expressed from the wild-type plasmid and the plasmid containing the c.449−9C>T variant were analyzed in the HEK293T cell line. We showed that both plasmids produce the same RT–PCR product corresponding to the normally spliced transcript (Figure 2A). Sanger sequencing also revealed no splicing changes. These results are consistent with the splicing program predictions. Thus, it can be assumed that the c.449−9C>T intron variant does not affect splicing.

In the same manner, we used the minigene splicing assay to determine the effect of the c.1177+5G>A variant on *SH3TC2* pre-mRNA splicing. For this purpose, we cloned three exons (10, 11, and 12) of the *SH3TC2* gene into a minigene construct. All these exons are located close to each other, and therefore splicing changes can affect all three exons. Indeed, the analysis of human ESTs and mRNAs has shown the existence of transcripts with retention of introns 10 and 11. Another feature of this locus is the unusually large length of exon 11 with a size of 1.7 kb.

Analysis performed on the HEK293T cell line revealed several different bands on agarose electrophoresis that were difficult to interpret. Therefore, we decided to carry out NGS sequencing of the obtained PCR products (Figure 2B). As a result, we found that intron 10 was partially retained in transfection with the WT plasmid. About 30% of all transcripts contain the whole intron 10. At the same time, the c.1177+5G>A variant disrupted the donor splicing site of intron 10, which resulted in the complete retention of intron 10 (Figure 2B). The length of this intron is 122 nucleotides. This insertion leads to a frameshift and the formation of a premature stop codon (NP_078853.2: p.Ala393GlyfsTer2). This change shortens the protein to less than one third of the wild-type length, which leads to its complete non-functionality. It has been experimentally established that 30% of the wild-type transcript is a transcript with intron 10 retention, which leads to the formation of a premature stop codon and most likely NMD. Obviously, 70% of the normal transcript is sufficient for the gene product to perform its functions. However, the presence of the c.1177+5G>A variant affects the splicing process, contributing to the retention of the intron in all transcripts of this allele, which leads to the absence of a functional RNA product.

## 3. Discussion

Here, we reported a family case of HMSN type 1 with peculiar molecular findings that required functional analysis of detected variants in two genes. Two affected siblings had clinical signs of peripheral neuropathy with significantly different severity of phenotypic features. Both patients had abnormal nerve conduction velocities corresponding with myelinopathy. NGS study revealed two single nucleotide variants in the *SH3TC2* gene and one earlier reported variant in the *MPZ* gene. Both genes are associated with HMSN type 1, and the detected variants could cause neuropathy in the proband and his sister. Due to the unavailability of the proband’s father, there were difficulties in conducting and interpreting the family segregation analysis of the identified variants.

The *MPZ* (NM_000530.8): c.449−9C>T variant frequency in the European population does not allow us to exclude it as a cause of autosomal dominant HMSN. It was not found in the proband’s mother, but it was present in both siblings. The father was unavailable, and his neurological status before his death was unknown, so we were unable to interpret the segregation analysis correctly. According to prediction programs, this intronic variant does not affect splicing, and it was confirmed by our functional analysis. Considering all the obtained data, we classified this variant as likely benign and did not regard it as the cause of the disease.

According to the segregation analysis results, the *SH3TC2* (NM_024577.4): c.279G>A (p.Lys93=) and c.1177+5G>A variants are most likely to be in a compound heterozygosity state. The effect of these variants on splicing (predicted by bioinformatic programs) was confirmed experimentally by functional analysis. In the current study, we show that the c.1177+5G>A variant leads to a frameshift and the formation of a premature termination codon NP_078853.2: p.Ala393GlyfsTer2, due to the insertion of 122 nucleotides caused by the retention of intron 10. This abnormal transcript is also observed when testing the wild-type minigene construct, but its proportion does not exceed 30% to compare with WT, while the c.1177+5G>A variant leads to the complete absence of a WT transcript. For the c.279G>A variant, the effect on splicing has been established: an insertion of 19 nucleotides of intron 3 with the formation of a premature TAA stop codon in position 127 of the amino acid sequence [11]. The frequencies of these variants also do not conflict with the recessive inheritance type of the disease.

Information about all the variants of the nucleotide sequence studied in this work was entered into the ClinVar NCBI database [12] (*MPZ* (NM_000530.8): c.449−9C>T - SUB12840206 and *SH3TC2* (NM_024577.4): c.1177+5G>A SUB12843611).

The clinical phenotype of the siblings corresponds with CMT4C, including the characteristic, although not specific for this form only [14,18], of spine deformation (observed in the brother). However, our patients have a later-than-usual manifestation of the disease: most studies mention the first decade of life [2,19], whereas the proband’s sister had minimal symptoms at the age of 25 years. Her brother had a manifestation in the second decade of life but had insignificant symptoms; therefore, he did not seek medical help, considering himself to be healthy until the age of 30 years. Such age of manifestation is described in the literature and can be explained by a milder effect of the mutation [20].

Determining the causative gene variant in this case was the key task. Variants in the *MPZ* gene have autosomal dominant inheritance, and if this variant was the cause of the neuropathy, the risk of affected offspring would be 50%. It should be noted that the risk of birth of a second affected child to a sick child exists even in healthy parents who are not carriers of the dominant pathogenic variant. This risk is due to the effect of germinal mosaicism [21,22]. The probability of having affected children for both patients is substantially lower if the causative gene is *SH3TC2* (with autosomal recessive inheritance), considering the rarity of the disorder, and could be lowered significantly by target analysis of the patients’ partners in case of planning pregnancy.

The implementation of NGS into everyday molecular genetic diagnostics of monogenic disorders provides a wide array of opportunities for detection of the molecular causes of genetically heterogeneous diseases. However, as of date, for many pathologies, the choice between variants in different genes showing signs of pathogenicity is a difficult task. This problem can be solved using family segregation analysis and functional analysis.

## 4. Materials and Methods

DNA was extracted from whole blood samples using a Wizard^®^ Genomic DNA Purification Kit (Promega, Madison, WI, USA) according to the manufacturer’s protocol. Quantitative analysis was carried out using the SALSA MLPA Probemix P405 (MRC-Holland, Amsterdam, The Netherlands). The proband’s DNA was analyzed using a custom AmpliSeq™ panel on an Ion Torrent S5 next generation sequencer. The panel includes coding gene sequences for *AARS1*, *BSCL2*, *EGR2*, *FIG4*, *GDAP1*, *GJB1*, *HINT1*, *HSPB1*, *INF2*, *LITAF*, *LRSAM1*, *MFN2*, *MME*, *MORC2*, *MPZ*, *NDRG1*, *NEFL*, *PMP22*, *PRX*, *SH3TC2*, and *SORD* genes. Sequencing results were analyzed using a standard automated algorithm for data analysis. The average coverage for this sample was 80× with the coverage width (20×) ≥ 90–94%. The detected variants were called according to the nomenclature presented on the http://varnomen.hgvs.org/recommendations/DNA website (accessed on 1 June 2023).

To assess the population frequencies of the identified variants, we used samples from the “1000 Genomes” project, ESP6500, and The Genome Aggregation Database v2.1.1. To evaluate the clinical significance of the identified variants, we used the OMIM database and the HGMD^®^ Professional database v2021.3. Assessment of the pathogenicity and causativity of genetic variants was carried out in accordance with the international recommendations for the interpretation of data obtained by massive parallel sequencing.

Automated Sanger sequencing was carried out using an ABIPrism 3500xl Genetic Analyzer (Applied Biosystems, Foster City, CA, USA) according to the manufacturer’s protocol. Primer sequences were chosen according to the NM_024577.4 reference sequence.

### Minigene Splicing Assay

To carry out the minigene splicing assay, we used a pSpl3-Flu2 vector (modification of the previously described pSpl3-Flu vector [23] with deletion of the part of the HIV tat intron containing a strong cryptic splice site). *MPZ* exon 4 with flanking intronic sequences were amplified using the patient’s genomic DNA as a template with the following primers: MPZ-XhoI-F1 (5′-AAAACTCGAGAAAGCCCGGCCTAAGGAC) and MPZ-BamHI-R1 (5′-AAAAGGATCCCTTGCACCGCGGACACAG). The PCR product was cloned into the pSpl3-Flu2 vector by XhoI/BamHI. The obtained wild-type (WT) and c.3140−16T>A plasmids were selected and verified by Sanger sequencing.

To create the minigene plasmids with the variant in the *SH3TC2* gene, a 2.2 kb region containing exons 10, 11, and 12 was cloned. The target locus was amplified using the patient’s genomic DNA as a template with the following primers: SH3TC2-BamHI-F1 (5′-AAAAGGATCCGGGGTACACCTGGAGGCA) and SH3TC2-NdeI-R1 (5′-AAAACATATGACCCTGACTCACCTGGCG). The PCR product was cloned into the pSpl3-Flu2 vector by BamHI/NdeI. The obtained wild-type (WT) and c.1177+5G>A plasmids were selected and verified by Sanger sequencing.

The minigene plasmids were separately transfected into HEK293T cells using the calcium phosphate method. After 48 h, the cells were harvested, and the total RNA was isolated using ExtractRNA reagent (Evrogen, Moscow, Russia) according to the manufacturer’s recommendations. Next, RNA samples were treated with DNaseI (Thermo Fisher Scientific, Waltham, MA, USA) and reverse-transcribed using a 5× RT MasMIX-30100 (Dialat Ltd., Moscow, Russia). To analyze the structure of the resultant chimeric transcript in the *MPZ* minigenes, we used the following plasmid-specific primers: TurboFP-F (5′-ACAAAGAGACCTACGTCGAGCA-3′) and GFP-R (5′-AGCTCGATCAGCACGGGCACGAT-3′). The PCR products were separated using 2% agarose gel electrophoresis. All amplicons were sequenced to verify the obtained transcript.

To analyze the structure of the transcripts expressed from the *SH3TC2* minigenes, we used a plasmid-specific primer TurboFP-F (5′-ACAAAGAGACCTACGTCGAGCA-3′) and a primer located in the *SH3CT2* exon 11 SH3CT2-R1 (5′-ACCCTGGCCTGAGAGAGTT-3′). The obtained PCR products were sequenced using NGS. NGS libraries were prepared and sequenced on an Ion Torrent S5 (with coverage of approximately 5000). The raw sequencing data was processed with a custom pipeline based on the open-source bioinformatics tools STAR 2.7.8a, Samtools, and SAJR. Splice junctions were visualized using the Sashimi plot in IGV.

This study was conducted according to the guidelines of the Declaration of Helsinki and approved by the Institutional Review Board of the Research Center for Medical Genetics, Moscow, Russia (approval number 2018-5/4). The probands gave informed consent to the genetic testing and the publication of anonymized data.

## 5. Conclusions

As a result of this work, the effect on splicing by retaining the intron of a previously undescribed variant *SH3TC2* (NM_024577.4):c.1177+5G>A and the absence of influence of a previously described variant *MPZ* (NM_000530.8):c.449−9C>T were shown.

## Figures and Tables

**Figure 1 ijms-24-09786-f001:**
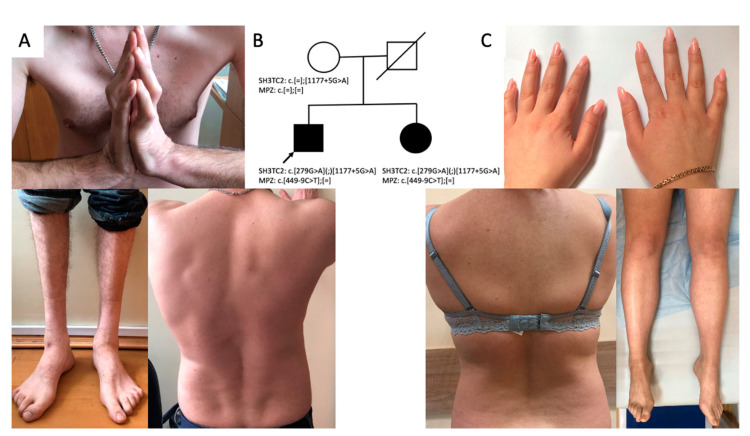
Clinical features of the proband: kyphoscoliosis, lower leg and hand muscle atrophy, hand joint contractures (**A**); genealogy (**B**); clinical features of the proband’s sister: mild hypotrophy of hand and foot muscles bilaterally, mild scoliosis (**C**).

**Figure 2 ijms-24-09786-f002:**
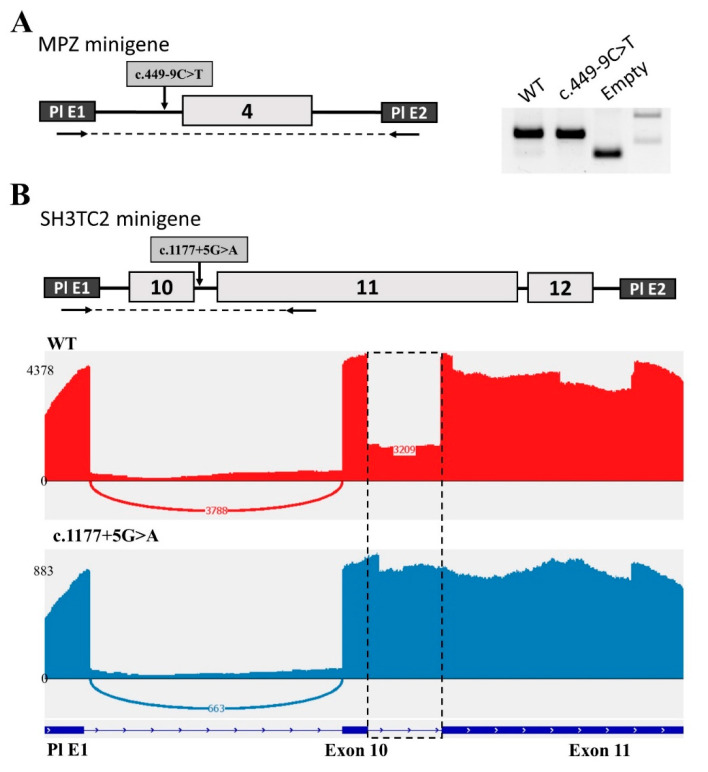
Minigene analysis of the c.449−9C>T variant in the *MPZ* gene and the c.1177+5G>A variant in SH3TC2. (**A**) Scheme of the *MPZ* minigene containing c.449−9C>T and electrophoregram of RT–PCR products. (**B**) Scheme of the SH3TC2 minigene containing c.1177+5G>A and Sashimi plots to visualize splice junctions for WT and c.1177+5G>A RT–PCR products of minigenes. Numbers represent the number of reads spanning the indicated exon/exon junction (EEJ).

## Data Availability

Raw data cannot be posted due to patient confidentiality. We can provide them or conduct a reanalysis upon request by mail schagina@med-gen.ru.

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
