# Peer review of "Evaluation of Pathogenicity and Causativity of Variants in the MPZ and SH3TC2 Genes in a Family Case of Hereditary Peripheral Neuropathy"

_ijms, 2023, doi:10.3390/ijms24129786_

Round 1

Reviewer 1 Report

The manuscript entitled “Evaluation of pathogenicity and causative of variants in the MPZ and SH3TC2 genes in a family case of hereditary peripheral neuropathy” by Elena Dadali and co-workers has shown the family case of HMSN type 1. By performing an NGS study they demonstrated that single nucleotide variants in the SH3TC2 and MPZ genes could cause neuropathy in the proband and his sister. Although, the initial observation is quite promising authors have not followed the observation further to understand the mechanistic details.  The manuscript is very descriptive and has minimal experimental evidence.

  I have no specific comments about the manuscript.   

Author Response

Dear reviewer!

Thank you for carefully reviewing our work.

The purpose of this study was to show how important and how difficult it is to choose a causative gene in the case of diagnosis of a genetically heterogeneous pathology.

In the described case, variants of both genes could explain polyneuropathy in siblings.

In order to select a gene, we investigated the effects exerted by the variants identified in patients at non-canonical splicing sites in the MPZ and SH3TC2 genes.

The RNA analysis was carried out because without it we lacked the criteria of pathogenic or benign. We have experimentally proved that the variant in the MPZ gene does not affect for the mRNA structure, and the previously undescribed variant in the SH3TC2 gene does.

Thanks to this, we were able to confirm the clinical diagnosis in siblings and determine the family genetics risks. This was the main task of the reported case.

We have tried to make additions to the text of the article that will allow us to understand the experiments more clearly.

Best regards

Olga

Reviewer 2 Report

Shchagina et al. have made a report on a family case of peripheral neuropathy (hereditary motor and sensory neuropathy; HMSN), of which there are many variants. They suggest that the most likely diagnosis is Charcot-Marie-Tooth 4C (CMT4C). It is not the diagnosis itself that is the most interesting part of the manuscript, but the molecular work that leads to likely mutational cause of the disease. There are a number of genes that are causally linked to CMT, and the authors have sequenced 21 of them, and further analysed their findings.

The manuscript is generally well written, and the language is good. Still, I think there are some points where the manuscript can be improved and made clearer to the reader as indicated below. The major problem is in the Results part, where one serious question arises. I do not see that the authors try to explain or discuss it.

Major issues:

The issue starts at l. 96 (and subsequent text): First, some revision could make it easier for the reader to follow the argumentation. On l. 97, it could be stated that the genetic analysis was done for all three persons. I would also add a sentence that explicitly state the three mutations found in the two siblings. In the present version, I first got a bit confused as the authors in the next paragraph (l. 100) jumped to a mutation in a Serbian person, and I wondered how both the mutation came into the picture. Additionally, by clearly stating the nature of the mutations here (at l. 99), and that the mutations were found in both siblings, it is easy to conclude that these mutations are unlikely de novo mutations that have happened independently for these two individuals. The subsequent conclusion is that they must have been inherited from the parents.

As MPS was used to sequence the genes, it should be easy to observe whether each of the mutation positions in SH3TC2 was homozygous or heterozygous in the two siblings. I don't think Ion Torrent gives long enough read length to cover both positions in SH3TC2 gene into one read, so as a start it is not possible to phase these mutations, telling whether they physically belong to the same allele or not. However, as stated by the authors, finding c.1177+5G>A in the mother, allows the conclusion that the two variants belong to different alleles, c.297G>A coming from the father and c.1177+5G>A coming from the mother. This is only clearly stated in the Discussion (l. 197), but that statement ought to come in the Results.

Shchagina et al. conclude that MPZ c.449-9C>T is likely benign (l. 155 and 195). Furthermore, SH3TC2 c.279G>A is synonymous and does not affect splicing (and is hence likely benign). On the other hand, they find a strong effect of SH3TC2 c.1177+5G>A. Here two questions arise:

(i) It is unclear to me what the authors mean by the sentences at l. 171-2.  Does the sentence in line 171-2 mean that the protein is shortened to 1/3 of the WT length? On l. 167 and 203, I interpret the 30% as the percentage of transcripts that contained the full intron, 70 % of transcripts contained parts of the intron, or were prematurely terminated, and none of the transcripts contained the normal (WT) sequence. Some reformulations are needed to make this clearer.

(ii) This is the most serious issue. If SH3TC2 c.1177+5G>A is the causal mutation, is there any particular reason why the mother is unaffected? Assuming that this is a recessive mutation, the mother (probably) has one normal and one affected allele, but the expression from the normal allele is enough to keep her non-symptomatic. Her children have received SH3TC2 c.279G>A from the father. As interpret the authors, this variant is synonymous, non-symptomatic and benign, and the protein is expressed in normal level from this allele.  Why are then the two siblings affected? They have a benign allele variant of MPZ (presumably heterozygous with the WT variant, which I suppose came from the mother); and they have two variant alleles of SH3TC2, one that is defect and one that gives a WT protein (as it is a synonymous variant). So why are the heterozygous siblings affected and the heterozygous mother non-affected? In other words, why is SH3TC2 c.1177+5G>A recessive in the mother, but not recessive in the siblings? This deserves some arguments and discussion from the authors. OK, it might be that I misunderstand, but then the authors need to explain this better.

Other and minor comments:

L. 10: The second sentence in the Abstract ("However,...") can be removed. Its meaning content is covered by the next two sentences.

L. 13: Suggestion: "Sequencing of 21 HMSN-associated genes detected two variants...." In the Abstract, it is not needed to tell that this was a custom panel, it is more important that it covered a range of genes.

L. 41-43: Reformulate? "....clinical correlation, diagnostic strategies are being replaced by a targeted next-generation sequencing as the most effective and economical approach."

L. 49-50: The formulations here could lead the uninitiated reader to believe that MPS is only used for panels, while it is also the most common method for WES (and WGS).

L. 52: ...undetected by targeted methods.

L. 57-62: Long and heavy sentence. One may interpret from the formulation that the authors knew which mutations they were looking for, while it is clear from the subsequent text that they did not. The diagnosis is here mentioned as CMT1, while later it is said that the diagnosis corresponds to CMT4C. Reformulate (just write that a person with HMSN symptoms, his sibling and his mother were investigated by sequencing 21 HMSN-related genes, and functional analyses were performed by the minigene construct of the three found mutations).

L. 69-73: Repetitive.

L. 89: sings -> signs

L. 186-188: Quite much can be concluded from the fact that both siblings have the same three variants, and the mother has only one of the three (see argument above). This also affects the formulation at l. 191-2.

L. 220-1: Parenthood – not a necessary piece of information for the reader, this is too personal.

L. 267: DNAse -> DNase

L. 278: Is coverage used in the same sense as on l. 235-236 (= sequencing depth), and the high coverage is reached because there is only one PCR product that is sequenced?

L 293-299: Remove last two sentences. A number of superfluous ".".

Author Response

We express our great gratitude to the reviewer for the thorough study of our manuscript and valuable comments. All (except one) comments were taken into account and eliminated.The step-by-step answer is in the attached file

Major issues:

The issue starts at l. 96 (and subsequent text): First, some revision could make it easier for the reader to follow the argumentation. On l. 97, it could be stated that the genetic analysis was done for all three persons. I would also add a sentence that explicitly state the three mutations found in the two siblings. In the present version, I first got a bit confused as the authors in the next paragraph (l. 100) jumped to a mutation in a Serbian person, and I wondered how both the mutation came into the picture. Additionally, by clearly stating the nature of the mutations here (at l. 99), and that the mutations were found in both siblings, it is easy to conclude that these mutations are unlikely de novo mutations that have happened independently for these two individuals. The subsequent conclusion is that they must have been inherited from the parents.

Explanatory sentences have been added to the text (yellow selection L.100-104).

As MPS was used to sequence the genes, it should be easy to observe whether each of the mutation positions in SH3TC2 was homozygous or heterozygous in the two siblings. I don't think Ion Torrent gives long enough read length to cover both positions in SH3TC2 gene into one read, so as a start it is not possible to phase these mutations, telling whether they physically belong to the same allele or not. However, as stated by the authors, finding c.1177+5G>A in the mother, allows the conclusion that the two variants belong to different alleles, c.297G>A coming from the father and c.1177+5G>A coming from the mother. This is only clearly stated in the Discussion (l. 197), but that statement ought to come in the Results.

You are absolutely right. Sequencing did not allow to phase mutations. Only the analysis of the mother allows us to assume exactly the compound-heterozygous state of the variants. Explanatory information added to the text. (yellow selection L.104-106)                               ).

Shchagina et al. conclude that MPZ c.449-9C>T is likely benign (l. 155 and 195). Furthermore, SH3TC2 c.279G>A is synonymous and does not affect splicing (and is hence likely benign). On the other hand, they find a strong effect of SH3TC2 c.1177+5G>A.

Here two questions arise:

(i) It is unclear to me what the authors mean by the sentences at l. 171-2.  Does the sentence in line 171-2 mean that the protein is shortened to 1/3 of the WT length? On l. 167 and 203, I interpret the 30% as the percentage of transcripts that contained the full intron, 70 % of transcripts contained parts of the intron, or were prematurely terminated, and none of the transcripts contained the normal (WT) sequence. Some reformulations are needed to make this clearer.

Explanations are included in the text of the manuscript  (yellow selection L.180-185)                               .

(ii) This is the most serious issue. If SH3TC2 c.1177+5G>A is the causal mutation, is there any particular reason why the mother is unaffected? Assuming that this is a recessive mutation, the mother (probably) has one normal and one affected allele, but the expression from the normal allele is enough to keep her non-symptomatic. Her children have received SH3TC2 c.279G>A from the father. As interpret the authors, this variant is synonymous, non-symptomatic and benign, and the protein is expressed in normal level from this allele.  Why are then the two siblings affected? They have a benign allele variant of MPZ (presumably heterozygous with the WT variant, which I suppose came from the mother); and they have two variant alleles of SH3TC2, one that is defect and one that gives a WT protein (as it is a synonymous variant). So why are the heterozygous siblings affected and the heterozygous mother non-affected? In other words, why is SH3TC2 c.1177+5G>A recessive in the mother, but not recessive in the siblings? This deserves some arguments and discussion from the authors. OK, it might be that I misunderstand, but then the authors need to explain this better.

A synonymous variant of the SH3TC2 gene was previously described in patients from the Czech Republic (Laššuthová P, Gregor M, Sarnová L, Machalová E, Sedláček R, Seeman P. Clinical, in silico, and experimental evidence for pathogenicity of two novel splice site mutations in the SH3TC2 gene. J Neurogenet. 2012 Sep;26(3-4):413-20. doi: 10.3109/01677063.2012.711398. Epub 2012 Sep 5. PMID: 22950825.). The authors conducted an RNA analysis that confirmed the effect of this nucleotide substitution on splicing predicted by the programs. We presented this information in the results section in lines 114-124.

Other and minor comments:

L 10: The second sentence in the Abstract ("However,...") can be removed. Its meaning content is covered by the next two sentences.

Thank you, we removed it

L 13: Suggestion: "Sequencing of 21 HMSN-associated genes detected two variants...." In the Abstract, it is not needed to tell that this was a custom panel, it is more important that it covered a range of genes.

The text has been changed (yellow selection L.12-13) 

L 41-43: Reformulate? "....clinical correlation, diagnostic strategies are being replaced by a targeted next-generation sequencing as the most effective and economical approach."

Correction made

L 49-50: The formulations here could lead the uninitiated reader to believe that MPS is only used for panels, while it is also the most common method for WES (and WGS).

The text has been changed (yellow selection L.48-50) 

L 52: ...undetected by targeted methods.

Rectified (yellow selection L.51)

L 57-62: Long and heavy sentence. One may interpret from the formulation that the authors knew which mutations they were looking for, while it is clear from the subsequent text that they did not. The diagnosis is here mentioned as CMT1, while later it is said that the diagnosis corresponds to CMT4C. Reformulate (just write that a person with HMSN symptoms, his sibling and his mother were investigated by sequencing 21 HMSN-related genes, and functional analyses were performed by the minigene construct of the three found mutations).

Changes have been made to the text (yellow selection L.57-61)

L 69-73: Repetitive.

Not replaced. Clinical information in this case was no less important for the assessment of pathogenicity than genetic

L 89: sings -> signs

Rectified (yellow selection L.88)

L 186-188: Quite much can be concluded from the fact that both siblings have the same three variants, and the mother has only one of the three (see argument above). This also affects the formulation at l. 191-2.

Yes, single-parent families for segregation are one of the big problems in our practice. Here we just wanted to "hint" at the need for extremely careful handling of segregation data

L 220-1: Parenthood – not a necessary piece of information for the reader, this is too personal.

Removed

L 267: DNAse -> DNase

Replaced

L 278: Is coverage used in the same sense as on l. 235-236 (= sequencing depth), and the high coverage is reached because there is only one PCR product that is sequenced?

Yes, that's exactly it. A deep coating of one product is necessary to see the small details of the possible impact.

L 293-299: Remove last two sentences. A number of superfluous ".".

Thank you. Deleted

Reviewer 3 Report

The comments are in attached file.

Author Response

We express our deep gratitude to the reviewer for an interesting discussion and competent comments. A huge request, if the reviewer knows the groups involved in the study of CMT modifying genetic factors, to connect us with them. We have a huge amount of material and we really want to work in this direction. Step-by-step responses to the review:

Two main questions need to be answered:

Why were only 3 individuals genetically tested. Are there siblings of the mother or father that can be tested? It is very surprising that the patient considered his condition normal. This suggests that those around him have the same health problem. A broader genetic and clinical examination of the family is necessary.

A very important note. We always collect as large a pedigree of our patients as possible. However, in this case it was not possible. The mother of the patients is the only child in the family, her parents are not alive. The family does not maintain contacts with the relatives of the patients' father after his departure. This is the main problem of geneticists - we want to work with as much pedigree and number of samples as possible, but we have to get the maximum information from what is in reality.

 A sister and brother represent dramatically different clinical symptoms when their genotypes are the same, at least in the panel studied. The phenomenon of gene interaction is known to result in the double mutant phenotype not being the sum of the phenotypes caused by each mutation individually.

This is a very important point. We will answer this question as follows: 1) for peripheral neuropathies, a broad intrafamilial heterogeneity of the severity of the phenotype is generally described. This cannot be explained only by age. Naturally, there is an influence of the products of some genes on others, especially considering that all proteins actually work in a complex to ensure a complex process of impulse transmission. In our opinion, the search for options that can aggravate or alleviate symptoms within the framework of this work will be speculation. To conduct such a study, representative samples of close relatives with different severity of IPN are needed. 2) Nevertheless, the nature of the variants of the SH3TC2 gene in this case allows for a simpler explanation. After all, splicing is a probabilistic process. Using the example of the SMN gene, a lot of factors that can influence it are described. And we often observe the greatest intrafamily heterogeneity precisely with mutations in non-canonical splice sites.

Why did the authors study a panel of 21 genes instead of 60-100 (as they claim is the norm)? It is very likely that the causative mutation is in a gene not tested or sister or brother have the additional modifying mutation.

For the diagnosis of genetically heterogeneous neuromuscular diseases, there is the best panel - this is exomic sequencing:) However, no one refuses to analyze the number of copies of PMP22 for diagnostics. As a rule, if a duplication is found as a result of the study, the study stops and a diagnosis is made. We have formed a panel of the most frequent genes not as an alternative to WES and WGS, but as a diagnostic stage, which is an alternative to Sanger sequencing of individual genes. We can look at a small panel quickly, cheaply, practically with the "eyes" of all genes without relying on bioinformatic algorithms. At the same time, the informative value of her research for the CMT diagnosis  is slightly less than 30%.

Moreover the text needs deep sorting out - there is too much discussion in the results section, e.g. lines 100-117, 130-147.

We agree with you that the arguments about pathogenicity could be described in the discussion section. However, the initial assessment of the pathogenicity of the identified variant according to the ACMП recommendations is an integral part of the molecular genetic analysis, therefore, we set out in the results section.

Additionally, the results and discussion section needs a few summarising sentences at the end, and the conclusion is a general statement, not the conclusion of the study.

The text has been changed (green selection in text)

The discussion should note the germline mosaicism for MPZ (Fabrizi et al, 2001, Neurology and Takashima et al, 1999, Neuromuscular Disorders).

Information and recommended links have been added to the discussion section (green selection in text).

Line 185 – The MPZ gene mutations could be also associated with HMSN type 2 – for review Acta Myol. 2004 May;23(1):6-9.

This information is very important, but does not affect the topics of this brief report. It was enough for us that mutations of both genes could be the cause of myelinopathy. As part of a short message, we do not want to overload the manuscript with information about the MPZ gene, which is extremely interesting from the point of view of various IPN phenotypes.

All gene names and names of transcripts should be in italics.

Eliminated

Minor corrections.

Line 25 distal amyotrophy is an archaism

Сorrected

Line 27 cranial nerve involvement is observed when this is acquired, not inherited.

The involvement of different pairs of cranial nerves has been described in various forms of CMT (Yu-ichi Noto et al. JNeurol Neurosurg Psychiatry 2015;86:378-384; Werheid F, Azzedine H, Zwerenz E, Bozkurt A, Moeller MJ, Lin L, Mull M, Häusler M, Schulz JB, Weis J, Claeys KG. Underestimated associated features in CMT neuropathies: clinical indicators for the causative gene? Brain Behav. 2016 Apr;6(4):e00451)

We observed similar features in our patients with mutations in the genes GJB1, MFN2 and GDAP1. Therefore, we have included this information in the introduction.

Line 31 some HMSN are axonopathies

Changed the wording

Line 32, 37, 220 genes do not cause diseases but mutations in them

corrected

 Lines 38-39 white matter alterations are very rare and non-specific

We have described this as a casuistry and would like to leave it in the text of the manuscript

 Line 43 what is “economical diagnostic strategy”?

Rephrased

Line 48 MLPA is not an abbreviation for quantitative analysis

Fixed.

 Line 49 What is massive parallel sequencing?

This is one of the synonyms of the NGS term. (https://en.wikipedia.org/wiki/Massive_parallel_sequencing)

Line 53 I disagree with the view. Doctors are very sceptical and VUS is VUS to them.

The final diagnosis is always at the discretion of the attending physician

Line 57, 242 What do the authors mean when they write pathogenicity and causality - what is the difference?

Pathogenicity is the effect of a variant on the splicing or functioning of the protein product of a gene. Causality - the ability of this variant to be the cause of a phenotype in a particular patient. For example, a 408 mutation in the PAH gene is certainly pathogenic, but it cannot be the cause of a phenotype other than hyperphenylalaninemia.

Round 2

Reviewer 2 Report

Shchagina et al. have revised their manuscript, somewhat improving some of their arguments, but at the expense of becoming more repetitive. Thus, I still think it is possible making the text clearer.

Most seriously, my major question from the initial review is still unclarified: What about the mother? In l. 102, it is stated: "She was the carrier of only one variant SH3C2:c1177+5G>A." I understand this as she was homozygous for this variant. If it had been clearly stated that she was heterozygous, with the other allele as WT, I wouldn't have any problems understanding the disease pattern in this little family. I had assumed heterozygosity in my original review, but it seems that I was wrong, which makes the question even more serious. Why isn't she affected, when this variant is traced to be non-functional in the siblings? It should have given the disease in the mother, and I cannot see that the authors give any reasonable explanation for why the mother is healthy.   

I don't think it helps to bring in genetic mosaicism (refs. 22, 23). I cannot see that the authors are able to explain how mosaicism is relevant either the mother or the siblings.

Minor comments:

L. 36: ...to suggest that the definite gene that pathogenic variants causing... -> ..gene that has pathogenic...

L. 47: MLPA is the abbreviation of...?

L. 53-54: Formulation. Is it a complication that variants may be detected?

L. 178: ...shortens the protein more than three times... -> ...shortens the protein to less than one third of wild type length...

L. 192: ...HSM type 1[space]with...

L. 236: "re-birth" doesn't sound good, "birth of a second affected child" is better.

Author Response

We once again express our gratitude to the reviewer for his attentiveness and excellent comments. All corrections have been made to the manuscript.

Most seriously, my major question from the initial review is still unclarified: What about the mother? In l. 102, it is stated: "She was the carrier of only one variant SH3C2:c1177+5G>A." I understand this as she was homozygous for this variant. If it had been clearly stated that she was heterozygous, with the other allele as WT, I wouldn't have any problems understanding the disease pattern in this little family. I had assumed heterozygosity in my original review, but it seems that I was wrong, which makes the question even more serious. Why isn't she affected, when this variant is traced to be non-functional in the siblings? It should have given the disease in the mother, and I cannot see that the authors give any reasonable explanation for why the mother is healthy.   

Thank you for your attention to detail.

We have added explanatory information to the text (L.102-103).

The mother is not affected because she is a heterozygous carrier of a variant in the recessive gene.

I don't think it helps to bring in genetic mosaicism (refs. 22, 23). I cannot see that the authors are able to explain how mosaicism is relevant either the mother or the siblings.

Thanks for the comment. Reviewer 3 believes that it is necessary to reflect everything, even the most unlikely variants of genotype distribution observed in this case.

Minor comments:

  1. 36: ...to suggest that the definite gene that pathogenic variants causing... -> ..gene that has pathogenic...

Thanks, fixed

  1. 47: MLPA is the abbreviation of...?

Thank you, decoded

  1. 53-54: Formulation. Is it a complication that variants may be detected?

That's exactly what we wanted to say. We might not find the right one then analyzing one gene. On the contrary, we find "not necessary" when analyzing many genes at the same time.

  1. 178: ...shortens the protein more than three times... -> ...shortens the protein to less than one third of wild type length...

Thanks, fixed

  1. 192: ...HSM type 1[space]with...

Thanks, fixed

  1. 236: "re-birth" doesn't sound good, "birth of a second affected child" is better.

Thanks, fixed

Reviewer 3 Report

The comments can be found in the attached file.

Author Response

Dear reviewer! Thank you for your careful study of our manuscript. 
The purpose of this work was to show that when switching to methods that provide a lot of information about the genetic sequence, we always face the problem of choosing a causal variant/s from a variety of VUS. The presented case visualizes this problem well - after all, even when examining a small panel, you can encounter this.
Our goal was not to study the causes of differences in the clinical features of patients and modifying genetics factors. Our task is to find the most probable cause of the disease.

Further my biggest objection to this work is the use of a very narrow panel of genes that have been studied. With the population and ethnic variability within the Russian Federation, it is difficult to expect much genetic homogeneity. Please provide statistical data on the frequency of different alleles of genes whose mutations lead to CMT diseases and show that those tested in the panel represent the TOP group, responsible for most of cases of these diseases. How this panel is related to the data presented by Cortese et al., (doi:10.1212/WNL.0000000000008672)

The panel used in the study is a diagnostic replacement for the sequencing of individual genes by Sanger. The set is formed from the NMSN genes that are frequent in most populations. Panels of different sizes are used for the diagnosis of neuropathy and the results of the research have been published in many papers.
Here https://doi.org/10.1002/ana.22166  the real clinical practice of a large genetic laboratory has been published and not all patients have had a genome-wide study. Researchers from Canada (http://dx.doi.org/10.1136/jmedgenet-2019-106641) and Norway  (DOI: 10.4045/tidsskr.14.1002) also used small panels for the study, achieving good diagnostic results. 
We are just now preparing for publication our large study on the genetic heterogeneity of CMT and related disorders in Russian patients. Our small study on the diagnosis of this disease using exome sequencing is published here https://doi.org/10.17650/2222-8721-2020-10-4. It should be noted that most of the patients studied here had undergone other studies prior to WES, and therefore it is impossible to present a general spectrum for this work. A lot of work on some frequent genes was published long before the advent of NGS Mersiyanova IV, Ismailov SM, Polyakov AV, et al. Screening for mutations in the peripheral myelin genes PMP22, MPZ and Cx32 (GJB1) in Russian Charcot-Marie-Tooth neuropathy patients [published correction appears in Hum Mutat 2000;16(2):175]. Hum Mutat. 2000;15(4):340-347. doi:10.1002/(SICI)1098-1004(200004)15:4<340::AID-HUMU6>3.0.CO;2-Y. Here are the Russian works on individual genes DOI: 10.1007/s11033-019-05238-z, https://doi.org/10.17650/2222-8721-2020-10-2-39-45, Milovidova TB, Dadali EL, Fedotov VP, Shchagina OA, Poliakov AV. Zh Nevrol Psikhiatr Im S S Korsakova. 2011;111(12):48-55. PMID: 22433810, Sharkova IV, Milovidova TB, Dadali EL, Poliakov AV. Zh Nevrol Psikhiatr Im S S Korsakova. 2012;112(7):42-47. PMID: 23011429
We have analyzed the experience of NGS in CMT from different countries. We made a small panel precisely as an alternative to Sanger sequencing, because we consider only WES to be a good panel for diagnostics at the next stage after analysis some frequent genes.
«In the cited paper, the authors show how, in the case of mutations in the SH3TC2 gene, the clinical picture still depends on the presence of a second mutation in one of the CMT genes - in their case, LITAF.»
The clinical polymorphism of hereditary diseases and its underlying causes is an interesting topic. Publications showing the weighting of the phenotype, new phenotypes [DOI:10.1001/archneur.63.1.112] or, conversely, soft phenotypes due to the carriage of mutations in different CMT genes have not yet been systematized. The descriptions are rather phenomenological and require further data accumulation and processing.

Moreover, it was shown that altered gene copy number including SH3TC2 can also be pathogenic. Thus, the authors should also provide in the manuscript the data about the copy number of the SH3TC2 gene in siblings.

CNVs of any gene, not only SH3TC2, can be the cause of the CMT phenotype. We analyzed the most frequent change - the number of copies of PMP22. The HGMD currently contains information about only one case of CNV as the cause of SH3TC2 neuropathy (CG200926). Based on this, we do not consider it appropriate to analyze the number of copies of the SH3TC2 gene in this particular case. After all, two variants have already been identified in probands, one of which was previously described as pathogenic, and we confirm the pathogenicity of the second in this study.

The limitation of the number of genes tested means that the diagnosis may be incorrect. Especially that the brother clinical manifestation is quite severe compared to the sister. Authors can’t also exclude that mutations are also hidden in noncoding regions of the patients genome. In such a case whole genome sequencing should be applied.

In this manuscript, we show that using even a small panel can give "too many" results. Genome research is certainly more informative. Further research in this case is possible only to try to find the genetic causes of the varying severity of the disease in siblings, which is not the task of this report.

Please do not use synonyms NGS and massive parallel sequencing in one text, stick to one. Even if I intuitively understand what massive parallel sequencing is, I have not encountered this term in scientific publications. I do not consider Wikipedia as a source of scientific knowledge and Wikipedia is not a platform for establishing correct scientific vocabulary.

Fixed. Replaced by NGS.

Lines 25-26 Please segregate into two separate sentences typical symptoms (to one, the first) and accompanying symptoms to the second. All symptomes cannot be listed one after the other without distinguishing their significance.

Thank you for your comment. Changes have been made to the text of the manuscript

Round 3

Reviewer 2 Report

Thank you for clarifying that the mother was heterozygous. Your previous formulation could only be understood as the mother was homozygous for the variant. May I suggest that you make this sentence even more clear: "She was found heterozygous for the SH3TC2 gene, carrying one normal allele and the variant SH3TC2(NM_024577.4):c.1177+5G>A."

Author Response

Dear reviewer!
Thank you for your comments and suggestions. The changes made have improved our manuscript.

Reviewer 3 Report

Dear authors,

I accept your responses.

Author Response

Thank you for an interesting discussion.